# Quantum anomalous Hall crystals in moiré bands with higher Chern number

**Raul Perea-Causin** ✉, **Hui Liu** ✉ & **Emil J. Bergholtz** ✉

The realization of fractional Chern insulators in moiré materials has sparked the search for further novel phases of matter in this platform. In particular, recent works have demonstrated the possibility of realizing quantum anomalous Hall crystals (QAHCs), which combine the zero-field quantum Hall effect with spontaneously broken discrete translation symmetry. Here, we employ exact diagonalization to demonstrate the existence of stable QAHCs arising from $\frac{2}{3}$-filled moiré bands with Chern number $C = 2$. Our calculations show that these topological crystals, which are characterized by a quantized Hall conductivity of 1 (in units of $e^2/h$) and a tripled unit cell, are robust in an ideal model of twisted bilayer-trilayer graphene—providing a novel explanation for experimental observations in this heterostructure. Furthermore, we predict that the QAHC remains robust in a realistic model of twisted double bilayer graphene and, in addition, we provide a range of optimal tuning parameters, namely twist angle and electric field, for experimentally realizing this phase. Overall, our work demonstrates the stability of QAHCs at odd-denominator filling of $C = 2$ bands, provides specific guidelines for future experiments, and establishes chiral multilayer graphene as a theoretical platform for studying topological phases beyond the Landau-level paradigm.

The rise of moiré materials in the last years has boosted the study of phases with intertwined many-body correlations and topology[1–3]. Concretely, fractional Chern insulators (FCIs)—lattice systems that exhibit the fractional quantum anomalous Hall effect due to spontaneous breaking of time-reversal symmetry[4–15]—were predicted[16–22] and later realized[23–28] in a series of moiré heterostructures based on either graphene or transition metal dichalcogenides. The demonstration of FCI phases in moiré superlattices generated widespread excitement over the possibility of achieving quantized Hall conductance, dissipationless edge currents, and anyonic excitations in an experimentally accessible and highly-tunable platform without the need for a magnetic field. More recently, most efforts have been directed to exploring phases that go beyond the conventional Laughlin and hierarchy fractional quantum Hall (FQH) states. On the one hand, many theoretical works[29–36] have proposed that FCI analogs to non-Abelian FQH states can be stabilized at moiré fractional fillings $\nu = \frac{1}{2}$[29–35], as well as $\nu = \frac{3}{5}, \frac{2}{5}$[36], with experimental signatures suggested to correspond to

the former[37]. On the other hand, a few works have recovered the concept of Hall crystal[38]—where the topology associated with the quantum Hall effect is accompanied by translation symmetry breaking—and predicted the emergence of these physics in different van der Waals heterostructures at zero magnetic field[39–50].

Recent studies have shed light on the emergence of anomalous Hall crystals, which break the continuous translation symmetry in a system with weak or absent moiré modulation[41–45] and where the Hall conductivity maintains an integer value throughout an extended continuous range of filling factors where the crystal remains stable—as observed experimentally in multilayer rhomboedral graphene[51]. Here, we focus on quantum anomalous Hall crystals (QAHCs), where the discrete translation symmetry in a moiré lattice is broken[40]—leading to an integer-quantized Hall conductivity appearing at fractional filling factors where a topological charge density wave (CDW) commensurate with the underlying moiré lattice can form. Such a mismatch between Hall conductivity and filling factor has been recently observed in

Department of Physics, Stockholm University, AlbaNova University Center, Stockholm, Sweden. ✉e-mail: raul.perea.causin@fysik.su.se; hui.liu@fysik.su.se; emil.bergholtz@fysik.su.se

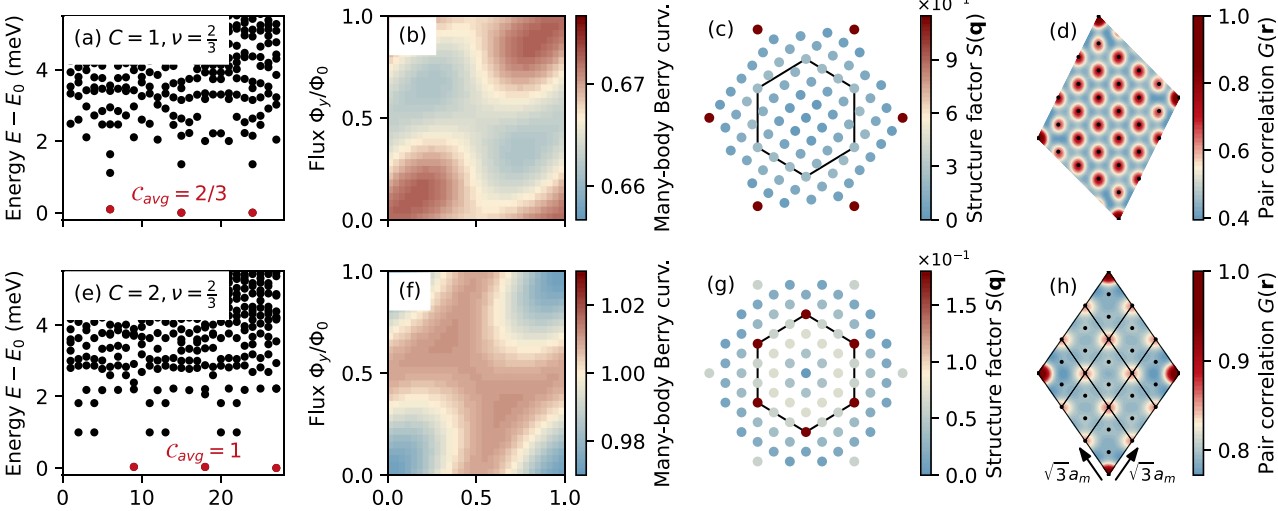

**Fig. 1 | Fractional Chern insulator (FCI) and quantum anomalous Hall crystal (QAHC) in ideal $C = 1$ and $C = 2$ bands. a** Many-body energy spectrum, **b** average Berry curvature, **c** structure factor, and **d** pair-correlation function demonstrating the FCI phase at $\nu = \frac{2}{3}$ filling of the ideal $C = 1$ band in a system with $N_s = 27$ sites and generating vectors $\mathbf{R}_1 = (6, 3)$, $\mathbf{R}_2 = (1, 5)$. **e–h** are the respective results for the ideal $C = 2$ band demonstrating an integer QAHC phase. In the latter case, the generating vectors are $\mathbf{R}_1 = (6, 3)$, $\mathbf{R}_2 = (3, 6)$. The Berry curvature $\Omega(\mathbf{k})$ is normalized by the number of flux points $N_\Phi = 30^2$, i.e., $\Omega(\mathbf{k})N_\Phi/2\pi$ is shown here so that the Chern number can be read off directly. The black dots in (**d**) and (**h**) denote moiré lattice sites, and the black lines in (**h**) mark the unit cells of the Hall crystal.

experiments on multilayer graphene systems[52–54]. However, despite the fact that the topological crystals presumably observed in bilayer-trilayer graphene might arise from a flat band with Chern number $C = 2$[53], previous theoretical works have been restricted to $C = 1$ bands. The stability of QAHCs arising from higher Chern bands is an important question that goes beyond the paradigm of traditional quantum Hall and Landau-level physics[55–57]. Moreover, numerical studies have so far only explored QAHCs at even-denominator filling, while experimental findings also point towards the existence of QAHCs at odd-denominator filling where this phase competes with FCIs and trivial CDWs.

In this work, we investigate the emergence of QAHCs in moiré bands with higher Chern number at odd-denominator filling factor, concretely $\nu = \frac{2}{3}$, by employing exact diagonalization (ED) of the many-body Hamiltonian. First, we consider ideal topological bands in the chiral model of twisted multilayer graphene[58,59]. Based on the degeneracy and Chern number of the many-body ground state as well as structure factor, pair-correlation function, and hole-entanglement spectrum (HES), we identify FCI, QAHC, and compressible liquid phases at $\frac{2}{3}$ filling of ideal $C = 1$, $C = 2$, and $C > 2$ moiré bands, respectively. In particular, the QAHC arising from the $C = 2$ band in the chiral model of twisted bilayer-trilayer graphene is characterized by an average many-body Chern number $\mathcal{C}_{\mathrm{avg}} = 1$, consistent with the experimental observations in this material[53], and a $\mathbf{K}$-point CDW modulation with a tripled unit cell corresponding to $\sqrt{3} \times \sqrt{3}$ moiré cells. Furthermore, we demonstrate that the QAHC arising from a $C = 2$ Chern band remains robust in a realistic model describing twisted double bilayer graphene (TDBG)[19,60–63]. Finally, we scan the parameter space of experimental tuning knobs and predict the optimal twist angles and layer potentials to realize this phase in TDBG.

## Results
### Ideal higher Chern bands
We consider ideal topological flat bands described by the chiral model of twisted multilayer graphene, which consists of two sheets of Bernal-stacked graphene, twisted by the magic angle and with artificially suppressed intra-sublattice tunneling between adjacent layers[58,59]. The top and bottom sheets consist of $n_t$ and $n_b$ layers, respectively, and the model hosts a pair of exact flat bands with Chern number $C = n_t$ and

$C = -n_b$, offering a perfect platform for investigating the physics of higher Chern bands. These bands have an ideal quantum geometry[58], i.e., the Fubini-Study metric is proportional to the Berry curvature distribution in momentum space, implying that FCIs are in principle the exact ground states of short-range pseudopotential interactions[64–70]. Here, however, we are interested in electrons interacting via the long-range Coulomb potential.

The simplest way to describe an ideal Chern band with $C = 1$ is by taking $n_t = n_b = 1$, corresponding to the chiral model of twisted bilayer graphene. This model has been used to emulate the lowest Landau-level physics[18] and to reveal the striking differences between such ideal bands and actual Landau levels[71]. Here, after solving the many-body problem projected onto the $C = 1$ band by ED (see "Methods"), we reveal a gapped phase characterized by three degenerate ground states appearing at the center-of-mass momenta expected for the $\nu = \frac{2}{3}$ FCI, cf. Fig. 1a. A more thorough analysis shows that the Berry curvature of the threefold degenerate many-body ground state is rather uniform [see Fig. 1b] and yields an average Chern number $\mathcal{C}_{\mathrm{avg}} = \frac{2}{3}$ for each ground state. The degeneracy and Chern number, as well as the HES (see details in the "Methods" section as well as Supplementary Note 1), indicate that the three ground states indeed correspond to the FCI phase. Aligned with this, the ground state structure factor $S(\mathbf{q})$ shown in Fig. 1c reveals a liquid-like feature, as no prominent peaks are observed in the moiré Brillouin zone (mBZ). The peaks outside the mBZ located at the $\Gamma$ points of the outer shell ($\mathbf{q} = C_6^n \mathbf{G}_1$, i.e., six-fold rotation $C_6^n$ of the reciprocal lattice vector $\mathbf{G}_1$) correspond to the first harmonics of the moiré potential. In fact, the real-space pair-correlation function $G(\mathbf{r})$ follows the periodic modulation of the moiré lattice [Fig. 1d]. For a definition of $S(\mathbf{q})$ and $G(\mathbf{r})$ we refer the reader to the "Methods" section.

We now explore the nature of the $\nu = \frac{2}{3}$ state in the ideal higher Chern band with $C = 2$. Motivated by the recent experimental realization of topological crystals in twisted bilayer-trilayer graphene[53], we set $n_t = 2$, $n_b = 3$ to model this heterostructure in the chiral limit. After performing ED on the $C = 2$ band with Coulomb interactions, we obtain a low-lying many-body energy spectrum that again exhibits a threefold degenerate ground state, see Fig. 1e. However, by relating to the generalized exclusion rule in the thin-torus limit[72,73], we rule out the competing order of a $\nu = \frac{2}{3}$ FCI phase, as then the three ground states

should share the same momentum (the $\Gamma$ point) for this specific tilted finite-size system; see details in Supplementary Note 2. Instead, the ground states are separated by the momentum $\mathbf{K}$ (or $\mathbf{K}'$). Interestingly, in this case the Berry curvature yields an average many-body Chern number $\mathcal{C}_{\text{avg}} = 1$ [Fig. 1f]. Based on previous works, the mismatch between the integer Chern number and the fractional filling factor $\nu$ suggests that the moiré translation symmetry is broken by the interactions. Indeed, a calculation of $S(\mathbf{q})$ confirms that the symmetry breaking is characterized by prominent peaks appearing at $\mathbf{K}$ and $\mathbf{K}'$ points [Fig. 1g], which implies a tripling of the unit cell. In Supplementary Fig. 2 we show that the intensity of $S(\mathbf{q} = \mathbf{K})$ increases with the system size, implying that the crystal is stable in the thermodynamic limit. A more intuitive picture can be obtained in real space, where the pair-correlation function $G(\mathbf{r})$ clearly follows a periodic crystal structure with a $\sqrt{3}a_m \times \sqrt{3}a_m$ unit cell ($a_m$ is the moiré lattice constant) that is three times larger than the moiré unit cell, cf. Fig. 1h. We emphasize that these results are in agreement with the observation of an integer-quantized Hall conductivity in twisted bilayer-trilayer graphene at $\nu = \frac{2}{3}$ filling[53]. Note that such an experimental signature has also been observed at $\nu = \frac{1}{3}$, although this phase seems to have a flipped Chern number—suggesting a different nature than the $\nu = \frac{2}{3}$ state potentially involving multi-valley physics. While we have considered the bilayer-trilayer structure in the ideal chiral limit, we will demonstrate later that the QAHC remains robust in a realistic model of twisted double bilayer graphene.

Interestingly, the absence of QAHC in the case of two degenerate ideal $C = 1$ bands (see Supplementary Fig. 3) suggests that the intrinsic higher-Chern number character of the $C = 2$ band is important for the stability of this phase. In fact, in a band with $C = 2$, Laughlin-like FCIs are expected to be most stable at $\nu = \frac{1}{5}$ and are absent at $\nu = \frac{1}{3}$[55,74]. Despite the lack of particle–hole symmetry in Chern bands, such absence of conventional FCI ordering might partially explain the robustness of QAHCs at the considered filling $\nu = \frac{2}{3}$. When considering the realistic model later on, we will show that the QAHC at $\nu = \frac{2}{3}$ is stabilized by an emergent kinetic energy of holes, while the quenched kinetic energy of flat-band electrons favors (Halperin-like) FCI order at $\nu = \frac{1}{3}$.

For even higher ideal Chern bands with $C > 2$, we have not observed any clearly gapped crystalline phase from the low-lying energy spectrum across various fillings. Instead, Fermi-liquid states dictated by an effective dispersion related to the fluctuating quantum metric dominate the phase diagram[75], see details in Supplementary Note 3. This is in line with results for another class of multilayer models with partially filled $C = n$ Chern bands, which cannot exhibit gapped spectra as $n \to \infty$[76]. We also note that the corresponding Wannier functions, bounded by the quantum metric, become extended for higher Chern number bands—making it more challenging to characterize any charge order[77]. On the other hand, the analogous situation in higher Landau levels with ideal quantum geometry $g_{\mathbf{k}} = (n + 1/2)\Omega_{\mathbf{k}}$, where the Hartree-Fock description of the fractional quantum Hall system becomes exact for large $n$[78,79], in principle allows for charge ordered states (but not FCIs).

We note that additional calculations show that these phases remain robust under variations in the number of layers in one sheet, e.g., $n_b$. This can already be seen from the fact that the single-particle quantum geometry of the target band remains unchanged and, consequently, considering interactions yields an identical behavior. Although this observation holds for the ideal chiral model, it prompts the question of to what extent it remains valid in realistic settings and whether it could aid future experimental investigations of such phases by simplifying the experimental setup, e.g., by considering bilayer-monolayer instead of bilayer-trilayer structures.

## Twisted double bilayer graphene
After having determined that QAHCs are stable at $\nu = \frac{2}{3}$ filling of an ideal flat band with Chern number $C = 2$, we demonstrate the

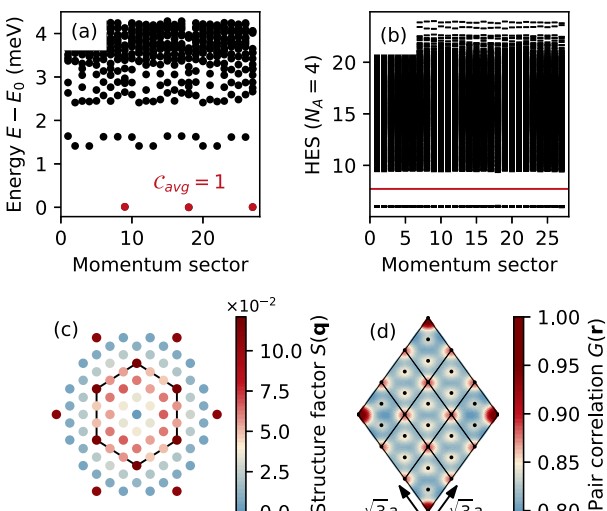

**Fig. 2 | Quantum anomalous Hall crystal at $\nu = \frac{2}{3}$ in realistic twisted double bilayer graphene with twist angle $\theta = 1.35°$ and a vertical potential bias of $U = 60$ meV. a** Many-body energy spectrum for $N_s = 27$ sites showing a threefold degenerate ground state with average many-body Chern number $\mathcal{C} = 1$. The parent single-particle band has a Chern number $C = 2$. **b** Hole-entanglement spectrum, where the number of states below the red line is 378 and matches exactly the number of allowed quasiparticle excitations in a charge density wave. **c** Structure factor in the moiré Brillouin zone. **d** Pair-correlation function in the considered finite system.

robustness of this phase in a realistic and experimentally accessible setting. In particular, we move away from the chiral limit and consider TDBG with finite intra-subband tunneling between adjacent layers, $w_0 = 0.7w_1$, where $w_1$ is the inter-sublattice tunneling strength[19]. This model is characterized by a $C = 2$ conduction band that remains isolated for a relatively broad range of twist angles and layer potentials[60,61], and which has been confirmed in experiments[80,81]. At filling $\nu = 1/3$ the conduction band has been predicted to harbor an FCI phase beyond the Landau-level paradigm[19].

We here consider the filling $\nu = 2/3$ and start with the twist angle $\theta = 1.35°$ and the layer potential $U = 60$ meV, at which the $C = 2$ Chern band is nearly flat and well isolated from neighboring bands[80]. Here, ED calculations with the same system size as above yield all the characteristic fingerprints of the QAHC phase: (i) a threefold degenerate many-body ground state with an average Chern number $\mathcal{C}_{\text{avg}} = 1$ [Fig. 2a]; (ii) a large gap in the HES, where the number of states below the gap is equal to the number of quasiparticle excitations in a CDW [Fig. 2b], see "Methods" and Supplementary Note 1 for more details; and (iii) a structure factor with pronounced $\mathbf{K}$-point peaks [Fig. 2c] resulting in a $\sqrt{3}a_m \times \sqrt{3}a_m$ CDW modulation [Fig. 2d]. In this realistic setting, though, the modulation arising from the moiré potential, which is visible in the strong $S(\mathbf{q})$ peaks at outer $\Gamma$ points in Fig. 2c, is quite significant. Nevertheless, we emphasize that even if the CDW modulation is weak, the QAHC phase can still remain robust as long as the moiré translation symmetry is broken.

Next, we aim to explore the robustness of the QAHC in TDBG across the $(U, \theta)$ parameter space and find the optimal values at which this phase is most stable. To this end, we perform ED on the $C = 2$ conduction band across the range $U \in [20\,\text{meV}, 100\,\text{meV}]$ and $\theta \in [1.1°, 1.6°]$ and extract the energy gap in the many-body spectrum between the QAHC ground and excited states. As shown in Fig. 3a, the QAHC remains stable with a gap ~1 meV for a relatively broad range of parameters, $U \in [50\,\text{meV}, 70\,\text{meV}]$ and $\theta \in [1.2°, 1.45°]$. A simple estimate relates these layer potentials to a displacement field $D = \epsilon U/d \in [0.18\,\text{V/nm}, 0.26\,\text{V/nm}]$, where we have taken $\epsilon = 5$ and $d = 4 \times 0.34\,\text{nm}$[82,83]. Importantly, the QAHC is stable in a parameter

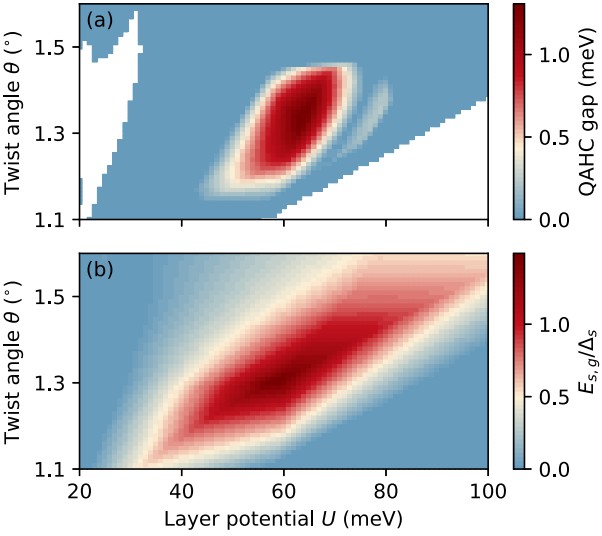

**Fig. 3 | Stability of the quantum anomalous Hall crystal (QAHC) in the ($U$, $\theta$) parameter space of twisted double bilayer graphene. a** Energy gap in the many-body spectrum with respect to the QAHC ground states. The regions where the single-particle band is not isolated appear in white. The many-body spectra have been generated for a system of $N_s = 21$ sites with generating vectors $\mathbf{R}_1 = (4, -1)$, $\mathbf{R}_2 = (1, 5)$. **b** Ratio between the energy gap $E_{s,g}$ and the bandwidth $\Delta_s$ of the single-particle band. The band has a Chern number $C = 2$ in the region where it is well isolated (see Supplementary Fig. 6).

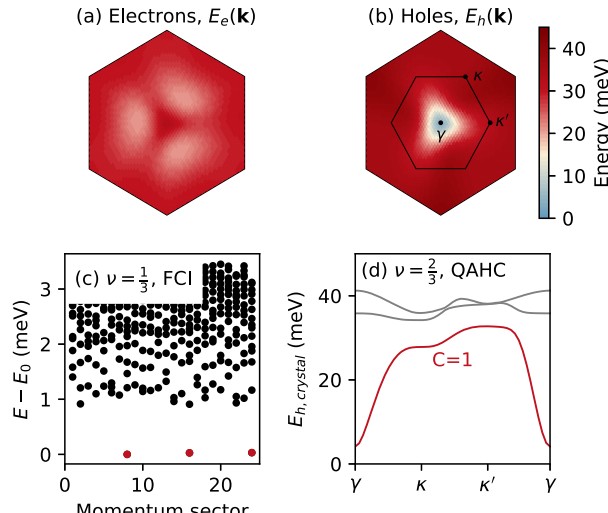

**Fig. 4 | Particle–hole symmetry breaking, fractional Chern insulator (FCI) at $\nu = \frac{1}{3}$ and quantum anomalous Hall crystal (QAHC) at $\nu = \frac{2}{3}$. a** Energy across the Brillouin zone for electrons in the empty band and **b** holes in the filled band of twisted double bilayer graphene. **c** Many-body spectrum at $\nu = \frac{1}{3}$ filling for $N_s = 24$ sites. The threefold degenerate ground states correspond to FCI order. **d** Band structure of holes perturbed by the potential $V_{\text{crystal}}(\mathbf{r})$. At filling $\nu = \frac{2}{3}$, the lowest band is fully occupied by holes, yielding an electron Chern number $C = -C_h = 1$ where $C_h$ is the Chern number of the hole band. The high-symmetry points $\gamma$, $\kappa$, and $\kappa'$ as well as the Brillouin zone of the QAHC are shown in panel (**b**).

region where the single-particle band is well isolated and flat, i.e., where the ratio between the single-particle band gap $E_{sp,g}$ and the bandwidth $\Delta_{sp}$ is maximized, cf. Fig. 3b. In addition, although there is an apparent closing and re-opening of the many-body gap at larger layer potentials, which is reminiscent of a phase transition, our calculations of HES, Chern number, and structure factor indicate that the fringe with a finite gap emerging at $U \approx 80$ meV still corresponds to the QAHC with $\mathcal{C}_{\text{avg}} = 1$ and **K**-point CDW modulation. Outside the optimal parameter region, the many-body energy gap closes and the system becomes a metal. We note that the QAHC phase persists when considering screening caused by metallic gates located at least $\approx 7$ nm away from the sample, and the phase diagram in Fig. 3a remains similar when considering layer-dependent interactions, see Supplementary Fig. 5.

The fact that the QAHC is most stable when the band is flat seems to oppose the recent observation that the stabilization of QAHCs requires a dispersive band[40]. However, this apparent contradiction can be resolved with the help of a particle–hole transformation. On the one hand, at filling $\nu = \frac{1}{3}$ the minority carriers are electrons which live in the nearly flat band that favors strongly-correlated FCI order[19], cf. Fig. 4a, c. On the other hand, the system at filling $\nu = \frac{2}{3}$ can be understood in terms of holes, where the band becomes renormalized due to the background electron–electron interactions[16] and, crucially, becomes dispersive, see Fig. 4b and "Methods" for details. Thus, the stabilization of the QAHC at $\nu = \frac{2}{3}$ can still be attributed to the (hole) band dispersion, which in the systems considered here arises exclusively due to the fluctuations of the quantum metric across the Brillouin zone[75]. We note that the lack of QAHC order in ideal Chern bands with $C \neq 2$ suggests that the quantum metric and the consequent hole dispersion must not just fluctuate but also have an appropriate distribution to be able to host charge order.

Finally, we show that the hole dispersion can be utilized to explain the topological nature of the QAHC from an effective single-particle description. We perturb the hole band with a potential that accommodates a $\sqrt{3} \times \sqrt{3}$ crystal, similar to the procedure followed in ref. 40. Concretely, we introduce the potential $V_{\text{crystal}}(\mathbf{r}) = 2V_0 \sum_n \cos(\mathbf{K}_n \cdot \mathbf{r})$, with $V_0 = -5$ meV, acting only on the two bottom layers. The resulting

band structure is characterized by three bands, see Fig. 4d, the lowest of which has a Chern number $C_h = -1$. The electron filling factor $\nu = \frac{2}{3}$ corresponds to filling the lowest band with holes, which yields an electron Chern number $C = -C_h = 1$, in agreement with the many-body calculations.

## Discussion

We have demonstrated the existence and robustness of QAHCs at $\nu = \frac{2}{3}$ filling of $C = 2$ Chern bands. First, we have shown that these phases emerge in ideal $C = 2$ bands, in particular in the chiral model of twisted bilayer-trilayer graphene—a heterostructure where experimental signatures of these topological crystals have been recently observed. Unlike the previously QAHCs predicted by numerical works, which were pinned at even-denominator filling factors, our results unveil topological crystals not only emerging at an odd-denominator filling factor $\nu = \frac{2}{3}$ but, moreover, originating from a higher Chern band with $C = 2$. This phase exhibits a **K**-CDW modulation, characterized by a $\sqrt{3}a_m \times \sqrt{3}a_m$ unit cell, and supports a quantized Hall conductance of 1 (in units of $e^2/h$). Following the finding of QAHCs in ideal $C = 2$ bands, we have demonstrated that this phase remains robust in a realistic setting, concretely in TDBG. Importantly, we have predicted that the QAHC in this heterostructure remains stable in a relatively wide range of experimentally accessible tuning parameters, namely for twist angles $\theta \in [1.2°, 1.45°]$ and layer potentials $U \in [50$ meV, $70$ meV$]$. Finally, we have provided a single-particle picture of the QAHC phase that reveals the important role of the non-flat hole dispersion—originating from quantum metric fluctuations—in stabilizing the crystal order.

From a theoretical perspective, the emergence of topological crystals in ideal flat bands establishes chiral twisted multilayer graphene as an excellent platform for further exploring novel properties of quantum anomalous Hall crystals and their connection to ideal quantum geometry. It has long been believed that the ideal quantum geometry favors the stabilization of FCIs against crystalline orders. However, the presence of QAHCs in this context challenges this general belief and therefore warrants further investigation. Additionally, as opposed to recent numerical studies on QAHCs in tMoTe$_2$ at even-

denominator filling factors, where the kinetic energy plays a dominant role in driving the crystal phase[40], our study suggests a contrasting scenario: only interaction matters, as the band dispersion in the chiral model is exactly flat. Importantly, our work also shows that in the present case, the QAHC stabilization can be connected to an effective (hole) kinetic energy that arises from interactions. Although twisted multilayer graphene differs substantially from moiré structures based on transition metal dichalcogenides, we believe our findings offer an alternative framework for understanding the fundamental nature of this new class of phases.

From a practical standpoint, our concrete prediction of a robust QAHC in TDBG considering experimentally accessible parameters offers a realistic and experimentally friendly guideline for discovering these phases. Given the strong connections of this realistic model to chiral models, our work could also be a starting point for searching for more exotic topological phases of matter in moiré materials within higher Chern bands, thus going beyond the traditional paradigm of Landau-level physics.

## Methods

### Exact diagonalization

To distinguish QAHCs and their competing orders, we employ exact diagonalization to numerically extract both the low-energy spectrum and ground-state information at fractional fillings $\nu = N_e/N_s$. Here, $N_e$ is the number of electrons, and $N_s$ is the number of moiré cells. The electron–electron interaction is projected onto the isolated and flat conduction band, and the resulting many-body Hamiltonian can be generally expressed as

$$H = \sum_{\mathbf{k}} E_{\mathbf{k}} c_{\mathbf{k}}^{\dagger} c_{\mathbf{k}} + \frac{1}{2} \sum_{\{\mathbf{k}_i\}} V_{\mathbf{k}_1 \mathbf{k}_2 \mathbf{k}_3 \mathbf{k}_4} c_{\mathbf{k}_1}^{\dagger} c_{\mathbf{k}_2}^{\dagger} c_{\mathbf{k}_3} c_{\mathbf{k}_4},$$

where $E_{\mathbf{k}}$ is the kinetic energy, $c_{\mathbf{k}}^{(\dagger)}$ is the electron annihilation (creation) operator with momentum $\mathbf{k}$, and $V_{\mathbf{k}_1 \mathbf{k}_2 \mathbf{k}_3 \mathbf{k}_4}$ is the Coulomb matrix element, which contains information about the single-particle wavefunction of the target band. We consider the Coulomb interaction $V(\mathbf{q}) = \frac{e_0^2}{2A\epsilon\epsilon_0|\mathbf{q}|}$ with the dielectric constant $\epsilon \approx 5$ that is typically considered in graphene systems[82].

### Structure factor and pair correlation

To further clarify the nature of the phases, we calculate the structure factor and pair-correlation function averaged over the threefold degenerate ground states, which provide insights into the crystalline or liquid character of the system. The structure factor is defined as $S(\mathbf{q}) = \frac{1}{N_e} \langle \rho(\mathbf{q})\rho(-\mathbf{q}) \rangle - N_e \delta_{\mathbf{q},0}$, where $\rho(\mathbf{q})$ is the density operator projected onto the considered flat band. On the other hand, the real-space pair-correlation function reads $G(\mathbf{r}) = \langle n(\mathbf{r})n(\mathbf{0}) \rangle$ up to a normalization factor, where $n(\mathbf{r})$ is the real-space density operator. We plot $S(\mathbf{q})$ with $\mathbf{q}$ only up to the closest reciprocal lattice vectors—we do not observe any significant peaks at larger $\mathbf{q}$, which in any case would simply correspond to spatial modulations of $G(\mathbf{r})$ at length scales shorter than the moiré lattice constant and would not affect the long-range CDW pattern.

### Entanglement spectrum

The fundamental nature of the many-body ground states can be accessed by the particle-cut entanglement spectrum (PES)[4,84]. The PES, not to be confused with entanglement entropy, is obtained by dividing the many-body system into $A$ and $B$ subsystems consisting of $N_A$ and $N_B = N_e - N_A$ particles, and then calculating the eigenvalues of $-\log \rho_A$, where $\rho_A = \text{tr}_B[\frac{1}{N_d} \sum_{i=1}^{N_d} |\Psi_i\rangle\langle\Psi_i|]$ is the reduced density matrix of $A$. Here, $N_d$ is the ground-state degeneracy. $\rho_A$ carries crucial information about the quasihole excitations in the degenerate ground states $|\Psi_i\rangle$, which are characteristically distinct for FCIs and CDWs. In particular, a gap in the PES is expected, and the number of states in the

lowest band of PES eigenvalues exactly matches the number of zero-energy quasihole excitations allowed by the specific quantum phase of the system. In this work, we have used the HES, where $N_A$ and $N_B = N_s - N_e - N_A$ now denote the number of holes[36]. The HES thus gives information about the quasiparticle excitations of the system.

### Hole energy

A particle–hole transformation of the band-projected many-body Hamiltonian leads to the hole energy

$$E_{\mathbf{k}}^{(h)} = -E_{\mathbf{k}} + E_{\mathbf{k}}^{(F)} - E_{\mathbf{k}}^{(H)},$$

which is renormalized due to the background electron–electron interactions of the filled band[16] emerging mainly through the term

$$E_{\mathbf{k}}^{(F)} = \sum_{\mathbf{q}} V(\mathbf{q}) |\langle \mathbf{k}+\mathbf{q}| e^{i\mathbf{q}\cdot\mathbf{r}} |\mathbf{k}\rangle|^2$$

and with a smaller contribution from

$$E_{\mathbf{k}}^{(H)} = \sum_{\mathbf{G}} V(\mathbf{G}) \langle \mathbf{k}| e^{i\mathbf{G}\cdot\mathbf{r}} |\mathbf{k}\rangle \sum_{\mathbf{k}'} \langle \mathbf{k}'| e^{-i\mathbf{G}\cdot\mathbf{r}} |\mathbf{k}'\rangle,$$

where $|\mathbf{k}\rangle$ is the single-particle Bloch state with momentum $\mathbf{k}$ for the considered band. The form factor $\mathcal{F}(\mathbf{k},\mathbf{q}) = \langle \mathbf{k}+\mathbf{q}| e^{i\mathbf{q}\cdot\mathbf{r}} |\mathbf{k}\rangle$ is related to the Fubini-Study metric $g_{ab}(\mathbf{k})$ via $|\mathcal{F}(\mathbf{k},\mathbf{q})|^2 \approx 1 - \sum_{a,b=x,y} q_a q_b g_{ab}(\mathbf{k})$ for small $\mathbf{q}$, showing that a $\mathbf{k}$-dependent $g_{ab}(\mathbf{k})$ results in a dispersive renormalized energy $E_{\mathbf{k}}^h$[75]. The hole dispersion in the TDBG band is shown in Fig. 3b.

## Data availability

All the data generated in this study are available in the article and supplementary information or from the corresponding authors upon request. Source data are provided with this paper.

## Code availability

The codes used to generate and analyze the data are available from the corresponding authors upon request.

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

## Acknowledgements

We acknowledge useful discussions and related collaborations with Zhao Liu, Ahmed Abouelkomsan, Liang Fu, Aidan Reddy, and Donna Sheng. This work was supported by the grants awarded to E.J.B. by the Swedish Research Council (2018-00313 and 2024-04567), the Knut and Alice Wallenberg Foundation (2018.0460 and 2023.0256), and the Göran Gustafsson Foundation for Research in Natural Sciences and Medicine. The computations were enabled by resources provided by the National Academic Infrastructure for Supercomputing in Sweden (NAISS), partially funded by the Swedish Research Council through grant agreement no. 2022-06725. In addition, we utilized the Sunrise HPC facility supported by the Technical Division of the Department of Physics, Stockholm University.

## Author contributions

R.P.C. and H.L. performed the numerical calculations. R.P.C., H.L., and E.J.B. conceived the research, analyzed and discussed the results, and wrote the manuscript.

## Funding

## Competing interests

The authors declare no competing interests.
