## [Transparent Peer Review file · Nature Communications]

Quantum anomalous Hall crystals in moiré bands with higher Chern number

Corresponding Author: Dr Raul Perea-Causin

Version 0:

Reviewer comments:

Reviewer #1

(Remarks to the Author)

This study demonstrates the existence of quantum anomalous Hall crystals (QAHCs) at $2/3$ filling of moiré bands with Chern number $C = 2$. Using exact diagonalization, the authors show that these topological crystals exhibit a quantized Hall conductivity of 1 (in units of e^2/h) and a tripled unit cell. The QAHC remains robust in ideal and realistic models of twisted bilayer-trilayer graphene and twisted double bilayer graphene (TDBG), respectively. For TDBG, they identify optimal experimental parameters: twist angles between 1.2° and 1.45° and layer potentials between 50-70 meV. This provides a novel explanation for recent experimental observations in multilayer graphene systems.

I believe that the work presented in this paper is a good candidate for Nature Communications. However, I am confused by some of the results and discussions in the manuscript. If the authors can address my questions satisfactorily, I will be happy to recommend it for publication in Nature Communications.

First, a significant gap in this paper is the lack of insight into the competition between FCI and QAHC phases in the $C=2$ band. While the authors convincingly rule out FCI phases at $\nu=2/3$ filling, they offer no explanation for why the same $C=2$ band in TDBG supports an FCI at $\nu=1/3$, as reported in their previous work [19]. This omission is particularly striking because understanding the filling-dependent phase competition could reveal fundamental principles governing topological phases in moiré materials. Though particle-hole symmetry is generally broken in flat bands (unlike in Landau levels), the authors should at minimum address this apparent contradiction. A comparative analysis of the energetics, quantum geometry, or interaction effects at these two fillings would substantially strengthen the paper and provide crucial context for their findings. For example, the $\nu=1/3$ filling can be viewed as a system of holes at $\nu=2/3$ filling (plus some corrections, of course). Why don't the holes form a QAHC phase? Why are the terms that break particle-hole symmetry so crucial that they kill the QAHC phase for holes at $\nu=2/3$ filling? Without such discussion, our understanding of what determines the ground state at different fractional fillings remains incomplete.

Second, the connection of the reported QAHC phase at $\nu=2/3$ filling and the quantum geometry of the parent $C = 2$ band is weak. While they presented the maximum fluctuation of the quantum metric in the Brillouin zone (BZ) of TDBG as a function of U and θ , it is unclear how quantum geometry affects the stability of the QAHC phase or the competition between the FCI and the QAHC phase for that matter. Are they suggesting that the QAHC phase in TDBG is almost unaffected by the quantum geometry of the band? Or do they think that the QAHC phase generally needs an ideal quantum geometry? In this regard, in Fig. 3, what do they think is the primary reason for the disappearance of the QAHC phase when they tune away from the optimal parameter range (twist angles between 1.2° and 1.45° and layer potentials between 50-70 meV)? Is it just that the bands become not-so-flat? If so, can they come up with a scenario where the QAHC phase is stable with a flat band but fluctuating quantum metric?

Third, how do the results change when the form of the interaction potential is modified? In this work, the authors only considered a long-range Coulomb potential $V(q)^{-1} = 2\epsilon_0\epsilon\epsilon_0|\vec{q}|^{-1}$ with $\epsilon = 5$. Do they think the phase diagram in TDBG will change if they modify the interaction potential by, say, including some short-range components or screening effects (such as multiplying the Coulomb potential with a $\tanh(qd)$, which is commonly used in the literature), or even making the interaction layer dependent?

Finally, this work raises an important theoretical question that remains unaddressed: how is a single-flavor $C=2$ band

fundamentally different from a spin-degenerate $C=1$ band? The topological and geometric properties of these two cases may appear similar in some respects (both having a total Chern number of 2), but they likely host different many-body states due to their distinct origins. Would the QAHC phase observed at $\nu = 2/3$ filling persist if calculations were performed in a spin-degenerate $C=1$ system instead? If the QAHC phase disappears in the latter case, it would suggest that the intrinsic higher-Chern character of the band—rather than just the total Chern number—plays a crucial role in stabilizing these topological crystalline states. This distinction could reveal profound insights about the interplay between band topology, internal degrees of freedom, and many-body correlations. Furthermore, experimental systems often feature spin degeneracy, making this comparison particularly relevant for predicting and interpreting experimental results. Exploring this distinction would strengthen the paper significantly and provide clearer guidance for experimental efforts to realize these exotic quantum phases.

Reviewer #2

(Remarks to the Author)
Please see PDF.

Version 1:

Reviewer comments:

Reviewer #1

(Remarks to the Author)

I thank the authors for carefully responding to my questions and also carrying out extensive new calculations. I can now recommend this manuscript for publication in Nature Communications.

Reviewer #2

(Remarks to the Author)

The authors have convincingly answered my questions and, I believe, those of the other referee. I recommend this work for publication.

One small comment: the authors may wish to remind the readers of the standard-but-perhaps-obscure fact that composite fermion theory applied to ideal bands predicts FCIs at fillings $\nu = 1/(2Cs+1)$. That is $\nu = 1/5$ (not $1/3!$) is the natural fillings for Laughlin FCIs in $C=2$ bands. From this perspective, the non-appearance of an FCI at $\nu=2/3$ in a $C=2$ band is relatively unsurprising, and QAHC or metals are natural stand-ins.

Response to the reviewers' comments

We thank the reviewers for carefully reading our manuscript and for providing valuable constructive feedback which helped us to further increase the clarity and impact of our manuscript. We provide a detailed point-by-point response to the comments below. Changes in the revised manuscript are marked in blue.

Reviewer: 1

- 1. Comment** “This study demonstrates the existence of quantum anomalous Hall crystals (QAHCs) at $2/3$ filling of moiré bands with Chern number $C = 2$. Using exact diagonalization, the authors show that these topological crystals exhibit a quantized Hall conductivity of 1 (in units of e^2/h) and a tripled unit cell. The QAHC remains robust in ideal and realistic models of twisted bilayer-trilayer graphene and twisted double bilayer graphene (TDBG), respectively. For TDBG, they identify optimal experimental parameters: twist angles between 1.2° and 1.45° and layer potentials between 50-70 meV. This provides a novel explanation for recent experimental observations in multilayer graphene systems.

I believe that the work presented in this paper is a good candidate for Nature Communications. However, I am confused by some of the results and discussions in the manuscript. If the authors can address my questions satisfactorily, I will be happy to recommend it for publication in Nature Communications.”

Answer We thank the referee for noticing the value of our work and for raising valuable questions.

- 2. Comment** First, a significant gap in this paper is the lack of insight into the competition between FCI and QAHC phases in the $C = 2$ band. While the authors convincingly rule out FCI phases at $\nu = 2/3$ filling, they offer no explanation for why the same $C = 2$ band in TDBG supports an FCI at $\nu = 1/3$, as reported in their previous work [19]. This omission is particularly striking because understanding the filling-dependent phase competition could reveal fundamental principles governing topological phases in moiré materials. Though particle-hole symmetry is generally broken in flat bands (unlike in Landau levels), the authors should at minimum address this apparent contradiction. A comparative analysis of the energetics, quantum geometry, or interaction effects at these two fillings would substantially strengthen the paper and provide crucial context for their findings. For example, the $\nu = 1/3$ filling can be viewed as a system of holes at $\nu = 2/3$ filling (plus some corrections, of course). Why don't the holes form a QAHC phase? Why are the terms that break particle-hole symmetry so crucial that they kill the QAHC phase for holes at $\nu = 2/3$ filling? Without such discussion, our understanding of what determines the ground state at different fractional fillings remains incomplete.

Answer We agree that the competition between FCI and QAHC phases deserves more attention. FCIs are expected to be more favorable in flat bands, while QAHCs (with integer quantization) generally require some dispersion [Phys. Rev. Lett. 133, 066601 (2024)]. As mentioned by the referee, we observed in a previous work that the nearly flat $C = 2$ band in TDBG hosts an FCI at $\nu = \frac{1}{3}$ filling. As also noted by the referee, the case of $\nu = \frac{2}{3}$ filling can be understood in terms of holes occupying $\frac{1}{3}$ of the full band. Importantly, the dispersion of holes in the full band is very different from that of electrons in the empty band—a fact that may be understood from the fluctuations of the quantum metric across the Brillouin zone [Phys. Rev. Res. 5, L012015 (2023)].

In the present system, the hole dispersion displays a much larger bandwidth (~ 40 meV) than the electron dispersion (~ 10 meV), see Fig. R1(a),(b). The moderate kinetic energy of holes is unfavorable for FCI order and, instead, favors charge order in the form of a QAHC.

Changes In order to shed light on the competition between the FCI and QAHC phases at $\nu = \frac{1}{3}$ and $\nu = \frac{2}{3}$, we have added a new figure (Fig. R1, new Fig. 4 in the main text) and discussion in the main manuscript. The new figure contains the dispersion for electrons in the empty band and for holes in the full band, the many-body spectrum for the $\nu = \frac{1}{3}$ FCI, and the band structure of holes forming a crystal with tripled unit cell, which provides a single-particle description of the QAHC at $\nu = \frac{2}{3}$. The new discussion reads:

The fact that the QAHC is most stable when the band is flat seems to oppose the recent observation that the stabilization of QAHCs requires a dispersive band. However, this apparent contradiction can be resolved with the help of a particle-hole transformation. On the one hand, at filling $\nu = \frac{1}{3}$ the minority carriers are electrons which live in the nearly flat band that favors strongly-correlated FCI order, cf. Fig. 4(a),(c). On the other hand, the system at filling $\nu = \frac{2}{3}$ can be understood in terms of holes, where the band becomes renormalized due to the background electron–electron interactions and, crucially, becomes dispersive, see Fig. 4(b) and Methods for details. Thus, the stabilization of the QAHC at $\nu = \frac{2}{3}$ can still be attributed to the (hole) band dispersion, which in the systems considered here arises exclusively due to the fluctuations of the quantum metric across the Brillouin zone. We note that the lack of QAHC order in ideal Chern bands with $C \neq 2$ suggests that the quantum metric and the consequent hole dispersion must not just fluctuate but also have an appropriate distribution to be able to host charge order.

We have also added the following details to the Methods section:

Hole energy

A particle-hole transformation of the band-projected many-body Hamiltonian leads to the hole energy

$$E_{\mathbf{k}}^{(h)} = -E_{\mathbf{k}} + E_{\mathbf{k}}^{(F)} - E_{\mathbf{k}}^{(H)},$$

which is renormalized due to the background electron-electron interactions of the filled band emerging mainly through the term

$$E_{\mathbf{k}}^{(F)} = \sum_{\mathbf{q}} V(\mathbf{q}) |\langle \mathbf{k} + \mathbf{q} | e^{i\mathbf{q}\cdot\mathbf{r}} | \mathbf{k} \rangle|^2$$

and with a smaller contribution from

$$E_{\mathbf{k}}^{(H)} = \sum_{\mathbf{G}} V(\mathbf{G}) \langle \mathbf{k} | e^{i\mathbf{G}\cdot\mathbf{r}} | \mathbf{k} \rangle \sum_{\mathbf{k}'} \langle \mathbf{k}' | e^{-i\mathbf{G}\cdot\mathbf{r}} | \mathbf{k}' \rangle,$$

where $|\mathbf{k}\rangle$ is the single-particle Bloch state with momentum \mathbf{k} for the considered band. The form factor $\mathcal{F}(\mathbf{k}, \mathbf{q}) = \langle \mathbf{k} + \mathbf{q} | e^{i\mathbf{q}\cdot\mathbf{r}} | \mathbf{k} \rangle$ is related to the Fubini-Study metric $g_{ab}(\mathbf{k})$ via $|\mathcal{F}(\mathbf{k}, \mathbf{q})|^2 \approx 1 - \sum_{a,b=x,y} q_a q_b g_{ab}(\mathbf{k})$ for small \mathbf{q} , showing that a \mathbf{k} -dependent $g_{ab}(\mathbf{k})$ results in a dispersive renormalized energy $E_{\mathbf{k}}^{(h)}$. The hole dispersion in the TDBG band is shown in Fig. 3(b).

3. Comment “Second, the connection of the reported QAHC phase at $\nu = 2/3$ filling and the quantum geometry of the parent $C = 2$ band is weak. While they presented the maximum fluctuation of the quantum metric in the Brillouin zone (BZ) of TDBG as a function of U and θ , it is unclear how quantum geometry affects the stability of the QAHC phase or the competition between the FCI and the QAHC phase for that

Figure R1: Energy across the Brillouin zone for (a) electrons in the empty band and (b) holes in the full band of twisted double bilayer graphene at $(U, \theta) = (60 \text{ meV}, 1.35^\circ)$. (c) Many-body spectrum at $\nu = \frac{1}{3}$ filling for $N_s = 24$ sites. The threefold degenerate ground states correspond to FCI order. (d) Band structure of holes perturbed by the potential $V_{\text{crystal}}(\mathbf{r})$, which breaks the moiré translation invariance and triples the unit cell. Filling $\nu = \frac{2}{3}$ corresponds to fully occupying the lowest band with holes, which yields an electron Chern number $C = 1$ in agreement with the many-body Chern number obtained for the QAHC. Note that $C = -C_h$ where C_h is the Chern number of the hole band. The high-symmetry points γ , κ , and κ' as well as the crystal Brillouin zone are shown in panel (b).

Figure R2: Hole band in TDBG for the parameters $(U, \theta) = (60 \text{ meV}, 1.35^\circ)$ in the optimal parameter region for a stable QAHC and $(U, \theta) = (40 \text{ meV}, 1.35^\circ), (90 \text{ meV}, 1.5^\circ)$ outside the QAHC region.

matter. Are they suggesting that the QAHC phase in TDBG is almost unaffected by the quantum geometry of the band? Or do they think that the QAHC phase generally needs an ideal quantum geometry? In this regard, in Fig. 3, what do they think is the primary reason for the disappearance of the QAHC phase when they tune away from the optimal parameter range (twist angles between 1.2° and 1.45° and layer potentials between 50-70 meV)? Is it just that the bands become not-so-flat? If so, can they come up with a scenario where the QAHC phase is stable with a flat band but fluctuating quantum metric?”

Answer As discussed in the previous point, the quantum geometry plays an important role in the stabilization of the QAHC phase. The fluctuating quantum metric turns the flat electron band into a dispersive band from the perspective of holes, see Fig. R1(a),(b). Since the only difference between the electron and hole pictures is that holes see a dispersive band instead of a flat band, the hole dispersion (and therefore the fluctuating quantum metric that originates it) must be responsible for the stabilization of the QAHC. In addition, the lack of QAHC order in ideal Chern bands with $C \neq 2$ suggests that the quantum metric and the consequent hole dispersion must not just fluctuate but also have an appropriate distribution to be able to host QAHC order. It is reasonable to argue that such distribution should be able to accommodate CDW instabilities.

Regarding the stability of the QAHC with respect to the external parameters, we note that the regime where this phase is stable overlaps with the parameter region where the electron band is most flat and isolated from neighboring bands (see Fig. 3 in the main manuscript). We emphasize that, in fact, the QAHC is stable in an exactly flat band with fluctuating quantum metric—which is precisely the case of the ideal $C = 2$ band studied in the first part of our work. A plausible explanation for the destruction of the QAHC phase outside the optimal parameter regime is that the hole dispersion resulting from the combination of a non-flat electron band and the fluctuating quantum metric does not enable the formation of charge order. However, we have investigated the distribution of the hole energy for different parameters and found that, while there are some changes that might hold the key to understanding the QAHC stability, no universally valid conclusions can be drawn directly from them (see Fig. R2). A more thorough understanding of the intricate interplay between the quantum geometry and energy dispersion and their role in the formation of QAHC order is worth pursuing in a future work. Finally, we note that the difficulty in addressing the stability of the QAHC by analyzing the quantum metric and electron or hole bands could simply be a symptom of the limitations of single-particle quantities in describing correlated many-body phenomena.

Changes The new figure and discussion mentioned in the previous comment emphasize the connection between the fluctuating quantum geometry, the hole dispersion, and the QAHC.

4. Comment “Third, how do the results change when the form of the interaction potential is modified? In this work, the authors only considered a long-range Coulomb potential $V(q)^{-1} = 2A\epsilon\epsilon_0|q|$ with $\epsilon = 5$. Do they think the phase diagram in TDBG will change if they modify the interaction potential by, say, including some short-range components or screening effects (such as multiplying the Coulomb potential with a $\tanh(qd)$, which is commonly used in the literature), or even making the interaction layer dependent?”

Answer Introducing a more realistic screening accounting for the presence of metallic gates, i.e. $V(q) = \frac{\epsilon_0^2}{2A\epsilon\epsilon_0q} \tanh(qd)$, is a good idea that should be helpful for the design of experiments. We have implemented this and observe that the phase remains stable if the gates are at least ≈ 7 nm away from the center of the graphene heterostructure,

Figure R3: (a) Energy gap between the highest QAHC many-body state and the lowest excited state as a function of the gate distance d considering the gate-induced interaction. Here the parameters $(U, \theta) = (60 \text{ meV}, 1.35^\circ)$ in the TDBG system are taken. (b) QAHC gap as a function of twist angle and layer potential considering layer-dependent interactions. In all calculations we considered $N_e = 14$ electrons in $N_s = 21$ sites.

see Fig. R3(a). Thus, the QAHC in TDBG should be robust in typical experimental setups, where the gate-sample distances are on the order of tens of nanometers. For shorter distances, the gap in the many-body energy spectrum closes and the system becomes a metal.

We have also considered the effect of making the interaction dependent on the layers where the two interacting electrons are located. Concretely, the potential between electrons in layers l and l' reads $V_{ll'}(q) = \frac{e_0^2}{2A\epsilon\epsilon_0q} \exp(-qd_g|l-l'|)$, where $d_g = 0.334 \text{ nm}$ is the separation between nearest graphene layers. The resulting many-body spectrum still corresponds to the QAHC. The phase diagram remains qualitatively the same, with the maximum many-body gap still being $\approx 1 \text{ meV}$, although the optimal parameter region has reduced slightly and shifted to somewhat larger layer potentials, cf. Fig. R3(b).

Changes We have added Fig. R3 to the Supplementary Information and the following discussion on the impact of gate-induced screening and layer-dependent interactions on our results:

We note that the QAHC phase persists when considering screening caused by metallic gates located at least $\approx 7 \text{ nm}$ away from the sample, and the phase diagram in Fig. 3(a) remains similar when considering layer-dependent interactions, see Supplementary Figure 5.

5. Comment “Finally, this work raises an important theoretical question that remains unaddressed: how is a single-flavor $C = 2$ band fundamentally different from a spin-degenerate $C=1$ band? The topological and geometric properties of these two cases may appear similar in some respects (both having a total Chern number of 2), but they likely host different many-body states due to their distinct origins. Would the QAHC phase observed at $\nu = 2/3$ filling persist if calculations were performed in a spin-degenerate $C = 1$ system instead? If the QAHC phase disappears in the latter case, it would suggest that the intrinsic higher-Chern character of the band—rather than just the total Chern number—plays a crucial role in stabilizing these topological crystalline states. This distinction could reveal profound insights about the interplay between band topology, internal degrees of freedom, and many-body correlations. Furthermore, experimental systems often feature spin degeneracy, making this comparison particularly relevant for predicting and interpreting experimental results. Exploring this distinction would strengthen the paper significantly and provide clearer guidance for experimental efforts to realize these exotic quantum phases.”

Figure R4: Many-body energy spectrum for $\nu = \frac{2}{3}$ filling of two degenerate ideal $C = 1$ bands encoded in the (a) total momentum sectors and (b) total pseudospin sectors computed in a system with $N_s = 15$ sites and $N_e = 20$ electrons. Here pseudospin refers to the band degree of freedom. The average many-body Chern number of each ground state is $\mathcal{C}_{\text{avg}} = \frac{4}{3}$ indicating FCI order. (c) Hole-cut entanglement spectrum containing 3250 states below the first entanglement gap denoted by the red line. The same state counting was obtained in the particle-cut entanglement spectrum of the FCI emerging at $\nu = \frac{1}{3}$ filling of the $C = 2$ band in TDBG.

Answer The referee raises a very interesting question which has motivated us to perform additional calculations. We have considered two degenerate ideal $C = 1$ bands arising from two copies of the chiral model of twisted bilayer graphene—a system that is in a way analogous to a quantum Hall bilayer. At filling $\nu = \frac{2}{3}$ we find a threefold degenerate many-body state with fractional many-body Chern number $\mathcal{C}_{\text{avg}} = \frac{4}{3}$ indicating FCI order (albeit with a very small energy gap), see Fig. R4(a). The total pseudospin of the ground states shows that these are unpolarized in the band degree of freedom, see Fig. R4(b). Moreover, the number of states below the first gap in the hole-cut entanglement spectrum [Fig. R4(c)] matches the quasi-hole excitation counting of the $\nu = \frac{1}{3}$ FCI in the $C = 2$ band of TDBG [Phys. Rev. Lett. 126, 026801 (2021)], which presumably corresponds to a Halperin (112) state [Phys. Rev. Res. 5, 023166 (2023)]. The presence of FCI order in the two degenerate $C = 1$ bands suggests that the intrinsic higher-Chern number character of the band is important for the stabilization of the QAHC phase. However, we note that a consistent comparison between the single $C = 2$ band and the two degenerate $C = 1$ bands is not straightforward since the quantum geometry distribution in the two systems is different.

Changes We have included the new Fig. R4 to the Supplementary Information and added the following sentence in the main manuscript:

Interestingly, the absence of QAHC in the case of two degenerate ideal $C = 1$ bands (see Supplementary Figure 3) suggests that the intrinsic higher-Chern number character of the band is important for the stability of this phase.

Reviewer: 2

- Comment** “In their manuscript ‘Quantum anomalous Hall crystals in moiré bands with higher Chern number’ the authors study the appearance of charge density wave order with quantized anomalous Hall conductivity from $C = 2$ Chern bands in both idealized and realistic systems. The key result is that, instead of the fractional Chern insulator one would expect at this filling in a $C = 1$ Chern band, such higher Chern bands tend to give charge density order (or Fermi liquids at even higher Chern numbers). This is an important and topical subject, and is directly relevant to several intriguing recent

experimental results, especially those from [52-53]. The results are presented in a clear and easily digestible way, with impeccable technical justification. Moreover, the phase diagrams in Fig. 3 should provide practical and immediately useful guidance for experimental groups.

I recommend this paper for publication after the authors address the minor issues noted below.”

Answer We are very grateful for the referee’s positive assessment and recommendation for publication after we address the referee’s insightful comments.

2. Comment “Independent from the scientific content of the paper, I want to note an issue of nomenclature. The term ‘quantum anomalous Hall crystal’, as used by the authors (following the literature [39-49]) refers to two different phases: (1) breaking of continuous translation symmetry \rightarrow discrete translation, yielding a Chern band and (2) breaking of discrete \rightarrow discrete translation symmetry, yielding a Chern band. In the topologically trivial case, one would call these phases ‘Wigner crystal’ and ‘charge density wave’ — distinct names for distinct phases. Not only is the symmetry content of these two topological phases different, but so are their experimental signatures: the first persists for a finite range of densities, while the second only appears at fixed rational densities. Moreover, both phases are extremely interesting (both inherently, in my opinion, and to the community), and are worthy of study in their own right and under their own names. I believe it is inappropriate and confusing to readers to conflate these phases.”

Answer We fully agree with the referee on this point. While we already make a distinction between phases where continuous or discrete translation symmetry is broken, we regrettably used the same name to refer to both. We will use the convention that part of the literature has taken, where quantum anomalous Hall crystal (QAHC) refers to phases with broken discrete translation symmetry akin to charge density waves or ‘generalized’ Wigner crystals [Phys. Rev. B 109, 115116 (2024), Phys. Rev. Lett. 133, 066601 (2024)], and the name anomalous Hall crystal (AHC) is reserved for the case where the continuous translation symmetry is broken as in a conventional Wigner crystal [Phys. Rev. Lett. 133, 206503 (2024), Phys. Rev. B 110, 205124 (2024), Phys. Rev. B 110, 205130 (2024), Phys. Rev. X 14, 041040 (2024)]. We also note that different names are being used in literature, e.g. QAHCs are also denoted topological crystals, topological charge density waves, or symmetry-broken Chern insulators, and AHCs are also referred to as extended anomalous quantum Hall states.

Changes We have adapted the text to clarify that QAHC here refers to the phases with broken discrete translation symmetry. The most important change is the following text introducing recent works on (quantum) anomalous Hall crystals:

Recent studies have shed light on the emergence of anomalous Hall crystals, which break the continuous translation symmetry in a system with weak or absent moiré modulation and where the Hall conductivity maintains an integer value throughout an extended continuous range of filling factors where the crystal remains stable—as observed experimentally in multilayer rhomboedral graphene. Here, we focus on quantum anomalous Hall crystals (QAHCs), where the discrete translation symmetry in a moiré lattice is broken—leading to an integer-quantized Hall conductivity appearing at fractional filling factors where a topological charge density wave (CDW) commensurate with the underlying moiré lattice can form. Such mismatch between Hall conductivity and filling factor has been recently observed in experiments on multilayer graphene systems.

3. Comment “I particularly thank the authors for carefully characterizing their phases within exact diagonalization, using spectral degeneracy, flux threading and the particle-entanglement

spectrum. Moreover, as moiré systems vary significantly over the Brillouin zone (and such variations are quite physically important [66]), it is necessary to study multiple system sizes to draw firm conclusions. I was happy to see the authors did exactly this in the supplement, and that the results were entirely consistent at different sizes. Taken together, this is a ‘gold standard’ use of exact diagonalization at a technical level.”

Answer We are very happy to read this comment. We indeed aimed to characterize these phases in an unambiguous, clear, and complete manner.

4. Comment “These bands have an ideal quantum geometry... [57].’ There is a large body of work studying ideal bands which the authors should cite so the reader can understand this field and its utility, such as the still-underappreciated early work of Roy and standard references including (but definitely not limited to) Wang et al [PRL 127 (2021)].”

Answer The referee is absolutely right. We have now added citations to the Roy [PRB 90, 165139 (2014)] and Wang [PRL 127 (2021)] as well as other works [PRB 108, 205144 (2023), PRL 114, 236802 (2015), SciPost Phys. 12, 118 (2022), PRB B 104, 115160 (2021), PRB 102, 165148 (2020)].

5. Comment “Almost all the topological phases (both fractional and CDW-like) present in ideal bands have simple Laughlin-like wavefunctions for pseudopotential interactions. Can the authors provide such an analytic understanding of the translation-breaking phase here? To my mind this is an important question because, as the authors already astutely note, in higher Landau levels one would expect quite different charge-density order such as stripes.”

Answer We would like to note that the QAHCs studied here involve integer quantization of the Hall resistance. Therefore, these phases can be understood in a single-particle picture and, in principle, do not require the use of Laughlin-like wave functions. Concretely, QAHCs can be modeled by a mean-field description that enforces the translation symmetry breaking [Phys. Rev. Lett. 133, 066601 (2024)].

Motivated by the referee’s comment, we have implemented an effective single-particle model to gain new insights into the nature of the QAHCs studied here. We consider the renormalized dispersion of holes in the filled band of TDBG, see Fig. R1(b), and introduce an external potential, $V_{\text{crystal}}(\mathbf{r}) = 2V_0 \sum_n \cos(\mathbf{K}_n \cdot \mathbf{r})$, with $V_0 = -5$ meV, which acts as a mean field that accommodates the $\sqrt{3} \times \sqrt{3}$ charge density wave. Forcing the potential to act only on the two bottom layers of the TDBG structure, we obtain a hole band structure where the lowest band has a Chern number $C_h = -1$, see Fig. R1(d). The electron filling factor $\nu = \frac{2}{3}$ corresponds to filling the lowest band with holes, thus obtaining the total Chern number $C = -C_h = 1$ in agreement with the many-body Chern number of the QAHC.

Changes We added Fig. R1, containing the effective single-particle model for the QAHC, to the main manuscript (Fig. 4). We also included the following discussion:

Finally, we show that the hole dispersion can be utilized to explain the topological nature of the QAHC from an effective single-particle description. We perturb the hole band with a potential that accommodates $\sqrt{3} \times \sqrt{3}$ crystal, similar to the procedure followed in Ref. [Phys. Rev. Lett. 133, 066601 (2024)]. Concretely, we introduce the potential $V_{\text{crystal}}(\mathbf{r}) = 2V_0 \sum_n \cos(\mathbf{K}_n \cdot \mathbf{r})$, with $V_0 = -5$ meV, acting only on the two bottom layers. The resulting band structure is characterized by three bands, the lowest of which has a Chern number $C_h = -1$. The electron filling factor $\nu = \frac{2}{3}$ corresponds to filling the lowest band with holes, which yields an electron Chern number

Figure R5: Structure factor over a longer range of momenta \mathbf{q} . The calculation has been done for the QAHC ground states in the ideal $C = 2$ band in a system with $N_s = 21$ sites at $\nu = \frac{2}{3}$ filling. The hexagons indicate the first Brillouin zone and the first outer shell of Brillouin zones.

$C = -C_h = 1$ in agreement with the many-body calculations.

- 6. Comment** “The momentum cutoff on the structure factor $S(q)$ in Fig 1(c,g) is rather small, ending at the location of the first non-trivial peak. To conclusively rule out other CDW patterns, the authors should show $S(q)$ over a broader range of momentum, (probably in the supplement).”

Answer We show in Fig. R5 that the structure factor at larger momenta is negligible. Weak peaks appear at the Γ points beyond the first shell but their intensity is not visible to the naked eye. We have checked even larger momenta than the ones showed in Fig. R5 and did not observe any additional peaks.

Note that structure factors $S(\mathbf{q})$ with \mathbf{q} further away than the shortest reciprocal lattice vectors would not affect the periodic pattern of a CDW with an enlarged unit cell. Instead, such peaks would correspond to spatial modulations of the pair-correlation function at length scales shorter than the moiré lattice constant.

Changes We have added the following clarification in the Methods section:

We plot $S(\mathbf{q})$ with \mathbf{q} only up to the closest reciprocal lattice vectors—we do not observe any significant peaks at larger \mathbf{q} , which in any case would simply correspond to spatial modulations of $G(\mathbf{r})$ at length scales shorter than the moiré lattice constant and would not affect the long-range CDW pattern.

- 7. Comment** “For Fig 3, the authors should characterize the competing phases to facilitate comparison with experiments. Presumably the competing phases are simply metals, but are other options possible such as a topologically trivial CDW? If I read their plots correctly (and take their model at face value), it seems that the twist angle in [52] is actually quite marginal to achieve the phase — perhaps explaining why the phase is so small there. Is this correct?”

Answer In principle, metals, trivial CDWs, and FCIs are competitors to the QAHC, but our exact diagonalization calculations show that QAHC is the stable phase at an optimal parameter region. In the TDBG phase diagram, we observe that outside the optimal parameter region the gap closes and the system becomes a metal.

We note a direct comparison between the phase diagram for twisted double bilayer graphene (TDBG) and the experiments performed on twisted trilayer-bilayer graphene should be done with care since these are slightly different systems. Nevertheless, it is reasonable to consider that the phase diagrams of these two systems could be similar, in which case it is true that the experimental twist angle $\theta = 1.5^\circ$ might be marginally large with respect to the optimal value for a robust QAHC. We would also like to clarify that it is the ideal model of trilayer-bilayer graphene (Fig. 1) which seems to provide an explanation for the experimental observation.

Changes We have added the following clarification in the main text:

Outside the optimal parameter region, the many-body energy gap closes and the system becomes a metal.

8. Comment “In the section on twisted double bilayer graphene, the authors have focused on $\nu = \frac{2}{3}$. However, the phase observed in [52] occurs only at $\nu = \frac{1}{3}$ but not at $\nu = \frac{2}{3}$. Given the admirable goal of the authors to ‘[offer] a realistic and experimentally friendly guideline’, this issue should be addressed. This is particularly interesting in light of the non-trivial kinetic energy of the band.”

Answer We would like to clarify that the experiments have been performed on twisted bilayer-trilayer graphene, so a comparison with our results on twisted double bilayer (i.e. twisted bilayer-bilayer) graphene should be done with care. We also point out that both fractions, $\nu = \frac{1}{3}$ and $\nu = \frac{2}{3}$, display experimental signatures of topological crystals at different displacement and magnetic fields, see Fig. 2 and Fig. 3 in Nature 637, 1084–1089 (2025). Note that, in the experiment, the $\nu = \frac{1}{3}$ signature seems to have a flipped Chern number (-1 instead of $+1$), suggesting a different—potentially more complex—nature than the $\nu = \frac{2}{3}$ state. Thus, we focus on the $\nu = \frac{2}{3}$ phase, which can be explained by our calculations. Investigating the origin of the $\nu = \frac{1}{3}$ phase in the trilayer-bilayer system is an interesting future research project that might require numerically-demanding calculations taking into account both K and K’ valleys.

Changes We have added the following sentence in the main text:

Note that such experimental signature has also been observed at $\nu = \frac{1}{3}$, although this phase seems to have a flipped Chern number—suggesting a different nature than the $\nu = \frac{2}{3}$ state potentially involving multi-valley physics.

9. Comment “In Fig 5 of the Supplement, the authors should specify the units for the quantum metric. For Supp. Fig 3, they may want to use a consistent colorscale to facilitate comparison between panels. Finally, the authors should consider adding a section to their Supplement giving the bandstructures of the models they consider to make the paper more self-contained.”

Answer We thank the referee for these suggestions. We now specify that the quantum metric shown is normalized by the Brillouin zone area, i.e. $g(\mathbf{k})A_{\text{BZ}}/2\pi$, and show the band structures in the new Supplementary Figure 8. Regarding the Berry curvature plots, using a unique range in the color scale would make the features of the $C = 1$ band barely visible, so we have decided to leave the figure unchanged.

We are very grateful for the referees feedback and for their final recommendation for publication. Regarding the suggestion of referee #2:

“One small comment: the authors may wish to remind the readers of the standard-but-perhaps-obscure fact that composite fermion theory applied to ideal bands predicts FCIs at fillings $\nu = 1/(2Cs+1)$. That is $\nu = 1/5$ (not $1/3!$) is the natural fillings for Laughlin FCIs in $C=2$ bands. From this perspective, the non-appearance of an FCI at $\nu=2/3$ in a $C=2$ band is relatively unsurprising, and QAHC or metals are natural stand-ins.”

We have added the following discussion in the main text (blue text in the manuscript pdf file):

In fact, in a band with $C = 2$, Laughlin-like FCIs are expected to be most stable at $\nu = 1/5$ and are absent at $\nu = 1/3$ [55, 74]. Despite the lack of particle-hole symmetry in Chern bands, such absence of conventional FCI ordering might partially explain the robustness of QAHCs at the considered filling, $\nu = 2/3$. When considering the realistic model later on, we will show that the QAHC at $\nu = 2/3$ is stabilized by an emergent kinetic energy of holes, while the quenched kinetic energy of flat-band electrons favors (Halperin-like) FCI order at $\nu = 1/3$.

In their manuscript “Quantum anomalous Hall crystals in moiré bands with higher Chern number” the authors study the appearance of charge density wave order with quantized anomalous Hall conductivity from $C = 2$ Chern bands in both idealized and realistic systems. The key result is that, instead of the fractional Chern insulator one would expect at this filling in a $C = 1$ Chern band, such higher Chern bands tend to give charge density order (or Fermi liquids at even higher Chern numbers). This is an important and topical subject, and is directly relevant to several intriguing recent experimental results, especially those from [52-53]. The results are presented in a clear and easily digestible way, with impeccable technical justification. Moreover, the phase diagrams in Fig. 3 should provide practical and immediately useful guidance for experimental groups.

I recommend this paper for publication after the authors address the minor issues noted below.

Independent from the scientific content of the paper, I want to note an issue of nomenclature. The term “quantum anomalous Hall crystal,” as used by the authors (following the literature [39-49]) refers to two different phases: (1) breaking of continuous translation symmetry \searrow discrete translation, yielding a Chern band and (2) breaking of discrete \searrow discrete translation symmetry, yielding a Chern band. In the topologically trivial case, one would call these phases “Wigner crystal” and “charge density wave” — distinct names for distinct phases. Not only is the symmetry content of these two topological phases different, but so are their experimental signatures: the first persists for a finite range of densities, while the second only appears at fixed rational densities. Moreover, both phases are extremely interesting (both inherently, in my opinion, and to the community), and are worthy of study in their own right and under their own names. I believe it is inappropriate and confusing to readers to conflate these phases.

Scientific Questions/Comments

1. I particularly thank the authors for carefully characterizing their phases within exact diagonalization, using spectral degeneracy, flux threading *and* the particle-entanglement spectrum. Moreover, as moiré systems vary significantly over the Brillouin zone (and such variations are quite physically important [66]), it is necessary to study multiple system sizes to draw firm conclusions. I was happy to see the authors did exactly this in the supplement, and that the results were entirely consistent at different sizes. Taken together, this is a “gold standard” use of exact diagonalization at a technical level.
2. “These bands have an ideal quantum geometry. . . [57].” There is a large body of work studying ideal bands which the authors should cite so the reader can understand this field and its utility, such as the still-underappreciated early work of Roy and standard references including (but definitely not limited to) Wang *et al* [PRL 127 (2021)].
3. Almost all the topological phases (both fractional and CDW-like) present in ideal bands have simple Laughlin-like wavefunctions for pseudopotential interactions. Can the authors provide such an analytic understanding of the translation-breaking phase here? To my mind this is an important question because, as the authors already

astutely note, in higher Landau levels one would expect quite different charge-density order such as stripes.

4. The momentum cutoff on the structure factor $S(\mathbf{q})$ in Fig 1(c,g) is rather small, ending at the location of the first non-trivial peak. To conclusively rule out other CDW patterns, the authors should show $S(\mathbf{q})$ over a broader range of momentum, (probably in the supplement).
5. For Fig 3, the authors should characterize the competing phases to facilitate comparison with experiments. Presumably the competing phases are simply metals, but are other options possible such as a topologically trivial CDW? If I read their plots correctly (and take their model at face value), it seems that the twist angle in [52] is actually quite marginal to achieve the phase — perhaps explaining why the phase is so small there. Is this correct?
6. In the section on twisted double bilayer graphene, the authors have focused on $\nu = \frac{2}{3}$. However, the phase observed in [52] occurs only at $\nu = \frac{1}{3}$ but not at $\nu = \frac{2}{3}$. Given the admirable goal of the authors to “[offer] a realistic and experimentally friendly guideline”, this issue should be addressed. This is particularly interesting in light of the non-trivial kinetic energy of the band.
7. In Fig 5 of the Supplement, the authors should specify the units for the quantum metric. For Supp. Fig 3, they may want to use a consistent colorscale to facilitate comparison between panels. Finally, the authors should consider adding a section to their Supplement giving the bandstructures of the models they consider to make the paper more self-contained.